# MULTI-SOURCE MULTI-VIEW TRANSFER LEARNING IN NEURAL TOPIC MODELING WITH PRETRAINED TOPIC AND WORD EMBEDDINGS

## ABSTRACT

Though word embeddings and topics are complementary representations, several past works have only used pretrained word embeddings in (neural) topic modeling to address data sparsity problem in short text or small collection of documents. However, no prior work has employed (pretrained latent) topics in transfer learning paradigm. In this paper, we propose a framework to perform transfer learning in neural topic modeling using (1) pretrained (latent) topics obtained from a large source corpus, and (2) pretrained word and topic embeddings jointly (i.e., multi-view) in order to improve topic quality, better deal with polysemy and data sparsity issues in a target corpus. In doing so, we first accumulate topics and word representations from one or many source corpora to build respective pools of pretrained topic (i.e., *TopicPool*) and word embeddings (i.e., *WordPool*). Then, we identify one or multiple relevant source domain(s) and take advantage of corresponding topics and word features via the respective pools to guide meaningful learning in the sparse target domain. We quantify the quality of topic and document representations via generalization (perplexity), interpretability (topic coherence) and information retrieval (IR) using short-text, long-text, small and large document collections from news and medical domains. We have demonstrated the state-of-the-art results on topic modeling with the proposed transfer learning approaches.

## 1 INTRODUCTION

Probabilistic topic models, such as LDA (Blei et al., 2003), Replicated Softmax (RSM) (Salakhutdinov & Hinton, 2009) and Document Neural Autoregressive Distribution Estimator (DocNADE) (Larochelle & Lauly, 2012) are often used to extract topics from text collections and learn latent document representations to perform natural language processing tasks, such as information retrieval (IR). Though they have been shown to be powerful in modeling large text corpora, the topic modeling (TM) still remains challenging especially in the sparse-data setting, especially for the cases where word co-occurrence data is insufficient e.g., on short text or a corpus of few documents. To this end, several works (Das et al., 2015; Nguyen et al., 2015; Gupta et al., 2019) have introduced external knowledge in traditional topic models via word embeddings Pennington et al. (2014). However, no prior work in topic modeling has employed topical embeddings (obtained from large document collection(s)), complementary to word embeddings.

*Local vs Global Views*: Though word embeddings (Pennington et al., 2014) and topics are complementary in how they represent the meaning, they are distinctive in how they learn from word occurrences observed in text corpora. Word embeddings have *local* context (*view*) in the sense that they are learned based on local collocation pattern in a text corpus, where the representation of each word either depends on a local context window (Mikolov et al., 2013) or is a function of its sentence(s) (Peters et al., 2018). Consequently, the word occurrences are modeled in a *fine-granularity*. On other hand, a topic (Blei et al., 2003; Gupta et al., 2019) has a *global* word context (*view*): TM infers topic distributions across documents in the corpus and assigns a topic to each word occurrence, where the assignment is equally dependent on all other words appearing in the same document. Therefore, it learns from word occurrences across documents and encodes a *coarse-granularity* description. Unlike topics, the word embeddings do not capture thematic structures (topical semantics) underlying in the document collection.

| Notation | Description | Notation | Description |
|---|---|---|---|
| LVT, GVT | Local-view Transfer, Global-view Transfer | $\mathbf{A}^k \in \mathbb{R}^{H \times H}$ | Topic-alignment in $\mathcal{T}$ and $\mathbf{Z}^k$ |
| MVT, MST | Multi-view Transfer, Multi-source Transfer | $K, D$ | Vocabulary size, document size |
| $\mathcal{T}, \mathcal{S}$ | A target domain, a set of source domains | $E, H$ | Word embedding dimension, #topics |
| $\lambda^k$ | Degree of relevance of $\mathbf{E}^k$ in $\mathcal{T}$ | $\mathbf{b} \in \mathbb{R}^K, \mathbf{c} \in \mathbb{R}^H$ | Visible-bias, hidden-bias |
| $\gamma^k$ | Degree of imitation of $\mathbf{Z}^k$ by $\mathbf{W}$ | $\mathbf{v}, k, \mathcal{L}$ | An input document, $k$th source, loss |
| $\mathbf{E}^k \in \mathbb{R}^{E \times K}$, | Word embeddings of $k$th source | $\mathbf{W} \in \mathbb{R}^{H \times K}$ | Encoding matrix of DocNADE in $\mathcal{T}$ |
| $\mathbf{Z}^k \in \mathbb{R}^{H \times K}$ | Topic embeddings of $k$th source | $\mathbf{U} \in \mathbb{R}^{K \times H}$ | Decoding matrix of DocNADE |

Table 1: Description of the notations used in this work

Consider the following topics ($Z_1$-$Z_4$), where ($Z_1$-$Z_3$) are respectively obtained from different (high-resource) source ($\mathcal{S}^1$-$\mathcal{S}^3$) domains whereas $Z_4$ from the (low-resource) target domain $\mathcal{T}$ in the data-sparsity setting:

$Z_1$ ($\mathcal{S}^1$): *profit, growth, stocks,* **apple**, *fall, consumer, buy, billion, shares* → *Trading*

$Z_2$($\mathcal{S}^2$): *smartphone, ipad,* **apple**, *app, iphone, devices, phone, tablet* → *Product Line*

$Z_3$ ($\mathcal{S}^3$): *microsoft, mac, linux, ibm, ios,* **apple**, *xp, windows* → *Operating System/Company*

$Z_4$ ($\mathcal{T}$): **apple**, *talk, computers, shares, disease, driver, electronics, profit, ios* → ?

Usually, the top words associated with topics learned on a large corpus are semantically coherent and represent meaningful semantics, e.g., *Trading, Product Line*, etc. However in sparse-data setting, topics (e.g., $Z_4$) are incoherent (*noisy*) and therefore, it is difficult to infer meaningful semantics. Additionally, notice that the word *apple* is topically/thematically contextualized (topic-word association) by different semantics in $\mathcal{S}^1$-$\mathcal{S}^3$ and referring to a *Company*.

Unlike the topics, word embeddings encode syntactic and semantic relatedness in fine-granularity and therefore, do not capture thematic structures. For instance, the top-5 nearest neighbors (NN) of *apple* (below) in the embeddings (Mikolov et al., 2013) space suggest that it refers to a *fruit*; however, they do not express anything about its thematic context, e.g., *Health*.

$$\textbf{apple} \stackrel{NN}{\Longrightarrow} \textit{apples, pear, fruit, berry, pears, strawberry}$$

**Motivation (1) Knowledge transfer using pretrained word and topic embeddings**: Essentially, the application of TM aims to discover hidden thematic structures (i.e., topics) in text collection; however, it is challenging in data sparsity settings, e.g, in a short and/or small collection. This leads to suboptimal text representations and incoherent topics (e.g., topic $Z_4$).

To alleviate the data sparsity issues, recent works (Das et al., 2015; Nguyen et al., 2015; Gupta et al., 2019) have shown that TM can be improved by introducing external knowledge, where they leverage pretrained word embeddings (i.e., local view) *only*. However, the word embeddings ignore the thematically contextualized structures (i.e., document-level semantics), and can not deal with ambiguity. Given that the word and topic representations encode complementary information, **no** prior work has explored transfer learning in TM using pretrained topics obtained from a large corpus.

**Motivation (2) Knowledge transfer from multiple sources of word and topic embeddings**: Knowledge transfer via word embeddings is vulnerable to negative transfer (Cao et al., 2010) on the target domain when domains are shifted and not handled properly. For instance, consider a short-text document $\mathbf{v}$: [`apple gained its US market shares`] in the target domain $\mathcal{T}$. Here, the word *apple* refers to a *company*, and hence the word vector of *apple* (about *fruit*) is an irrelevant source of prior knowledge for both $\mathbf{v}$ and the topic $Z_4$. In contrast, one can better model $\mathbf{v}$ and amend the noisy $Z_4$ for coherence, given the meaningful word and topic embeddings.

Often, there are several topic-word associations in different domains, e.g., in topics $Z_1$-$Z_3$. Given a noisy topic $Z_4$ in $\mathcal{T}$ and meaningful topics $Z_1$-$Z_3$ of $\mathcal{S}^1$-$\mathcal{S}^3$, we identify multiple relevant (source) domains and advantageously transfer their word and topic embeddings in order to facilitate meaningful and positive transfer learning in the sparse corpus, $\mathcal{T}$.

**Contribution (1)** To our knowledge, it is the *first work* in unsupervised topic modeling framework that introduces (external) knowledge transfer using (a) *Global-view Transfer*: Pretrained topic em-

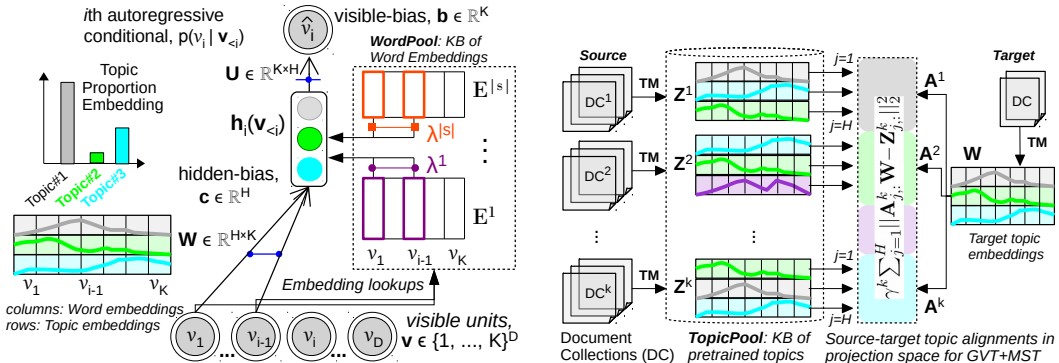

Figure 1: (Left) DocNADE (LVT+MST): Multi-source transfer learning in TM by introducing pretrained word embeddings from a *WordPool* at each autoregressive step $i$. Double circle $\rightarrow$ multinomial (softmax) unit. (Right) Multi-source transfer learning in TM by introducing pretrained (latent) topic embeddings from a *TopicPool*, illustrating topic alignments between source and target corpora in GVT+MST configuration. Each outgoing row from $\mathbf{Z}^k$ signify a topic embedding of the corresponding source corpus, $DC^k$. Here, TM refers to a DocNADE topic model.

beddings instead of using word embeddings exclusively, and (b) *Multi-view Transfer*: Pretrained word and topic embeddings jointly obtained from a large source corpus in order to deal with polysemy and alleviate data sparsity issues in a small target corpus.

**Contribution (2)** *Multi-source Transfer*: Moreover, we first learn word and topic representations on multiple source domains to build *WordPool* and *TopicPool*, respectively and then perform *multi-view* and *multi-source* transfer learning within neural topic modeling by jointly using the complementary representations. In doing so, we guide the (unsupervised) generative process of learning hidden topics of the target domain by embeddings in *WordPool* and *TopicPool* such that the hidden topics become more meaningful and representative in explaining the target corpus.

We evaluate the effectiveness of our transfer learning approaches in neural topic modeling using 7 (5 low-resource and 2 high-resource) target and 5 (high-resource) source corpora from news and medical domains, consisting of short-text, long-text, small and large document collections. Particularly, we quantify the quality of text representations via generalization (perplexity), interpretability (topic coherence) and text retrieval. *The code is provided with the supplementary.*

## 2 KNOWLEDGE TRANSFER IN NEURAL TOPIC MODELING

Consider a sparse target domain $\mathcal{T}$ and a set of $|\mathcal{S}|$ source domains $\mathcal{S}$, we first prepare two knowledge bases (KBs) of representations (or embeddings) from each of the sources: (1) *WordPool*: Pretrained word embeddings matrices $\{\mathbf{E}^1, ..., \mathbf{E}^{|\mathcal{S}|}\}$, where $\mathbf{E}^k \in \mathbb{R}^{E \times K}$, and (2) *TopicPool*: Pretrained latent topic embeddings $\{\mathbf{Z}^1, ..., \mathbf{Z}^{|\mathcal{S}|}\}$, where $\mathbf{Z}^k \in \mathbb{R}^{H \times K}$ encodes a distribution over a vocabulary of $K$ words. $E$ and $H$ are word embedding and latent topic dimensions, respectively. While topic modeling on $\mathcal{T}$, we introduce the two types of knowledge transfers from one or many sources: *Local* (LVT) and *Global* (GVT) *View Transfer* using the two KBs of pretrained word (i.e., *WordPool*) and topic (i.e., *TopicPool*) embeddings, respectively. Specially, we employ a neural autoregressive topic model (i.e., DocNADE (Larochelle & Lauly, 2012)) to build the *WordPool* and *TopicPool*.

*Notice* that a superscript indicates a source. See Table 1 for the notations used in this work.

### 2.1 NEURAL AUTOREGRESSIVE TOPIC MODELS

DocNADE (Larochelle & Lauly, 2012) is an unsupervised neural-network based topic model that is inspired by the benefits of NADE (Larochelle & Murray, 2011) and RSM (Salakhutdinov & Hinton, 2009) architectures. RSM has difficulties due to intractability leading to approximate gradients of the negative log-likelihood, while NADE does not require such approximations. On other hand, RSM is a generative model of word count, while NADE is limited to binary data. Specifically, Doc-

NADE factorizes the joint probability distribution of words in a document as a product of conditional distributions and efficiently models each conditional via a feed-forward neural network.

---

**Algorithm 1** Computation of $\log p(\mathbf{v})$ and Loss $\mathcal{L}(\mathbf{v})$

---

> **Input**: A target training document $\mathbf{v}$, $|\mathcal{S}|$ source domains
> **Input**: *WordPool*: KB of pretrained word embedding matrices $\{\mathbf{E}^1, ..., \mathbf{E}^{|\mathcal{S}|}\}$
> **Input**: *TopicPool*: KB of pretrained latent topics $\{\mathbf{Z}^1, ..., \mathbf{Z}^{|\mathcal{S}|}\}$
> **Parameters**: $\mathbf{\Theta} = \{\mathbf{b}, \mathbf{c}, \mathbf{W}, \mathbf{U}, \mathbf{A}^1, ..., \mathbf{A}^{|\mathcal{S}|}\}$
> **Hyper-parameters**: $\theta = \{\lambda^1, ..., \lambda^{|\mathcal{S}|}, \gamma^1, ..., \gamma^{|\mathcal{S}|}, H\}$
> Initialize: $\mathbf{a} \leftarrow \mathbf{c}$ and $p(\mathbf{v}) \leftarrow 1$
> **for** $i$ from 1 to $D$ **do**
>     $\mathbf{h}_i(\mathbf{v}_{<i}) \leftarrow g(\mathbf{a})$, where $g = \{\text{sigmoid, tanh}\}$
>     $p(v_i = w|\mathbf{v}_{<i}) \leftarrow \frac{\exp(b_w + \mathbf{U}_{w,:}\mathbf{h}_i(\mathbf{v}_{<i}))}{\sum_{w'} \exp(b_{w'} + \mathbf{U}_{w',:}\mathbf{h}_i(\mathbf{v}_{<i}))}$
>     $p(\mathbf{v}) \leftarrow p(\mathbf{v})p(v_i|\mathbf{v}_{<i})$
>     compute pre-activation at step, $i$: $\mathbf{a} \leftarrow \mathbf{a} + \mathbf{W}_{:,v_i}$
>     **if** `LVT` **then**
>         get word embedding for $v_i$ from source domain(s)
>         $\mathbf{a} \leftarrow \mathbf{a} + \sum_{k=1}^{|\mathcal{S}|} \lambda^k \, \mathbf{E}_{:,v_i}^k$
> $\mathcal{L}(\mathbf{v}) \leftarrow -\log p(\mathbf{v})$
> **if** `GVT` **then**
>     $\mathcal{L}(\mathbf{v}) \leftarrow \mathcal{L}(\mathbf{v}) + \sum_{k=1}^{|\mathcal{S}|} \gamma^k \sum_{j=1}^{H} ||\mathbf{A}_{j,:}^k \mathbf{W} - \mathbf{Z}_{j,:}^k||_2^2$

---

**DocNADE Formulation**: For a document $\mathbf{v} = (v_1, ..., v_D)$ of size $D$, each word index $v_i$ takes value in $\{1, ..., K\}$ of vocabulary size $K$. DocNADE learns topics in a language modeling fashion (Bengio et al., 2003) and decomposes the joint distribution $p(\mathbf{v}) = \prod_{i=1}^{D} p(v_i|\mathbf{v}_{<i})$ such that each autoregressive conditional $p(v_i|\mathbf{v}_{<i})$ is modeled by a feed-forward neural network using preceding words $\mathbf{v}_{<i}$ in the sequence:

$$\mathbf{h}_i(\mathbf{v}_{<i}) = g(\mathbf{c} + \sum_{q<i} \mathbf{W}_{:,v_q}) \quad \text{and} \quad p(v_i = w|\mathbf{v}_{<i}) = \frac{\exp(b_w + \mathbf{U}_{w,:}\mathbf{h}_i(\mathbf{v}_{<i}))}{\sum_{w'} \exp(b_{w'} + \mathbf{U}_{w',:}\mathbf{h}_i(\mathbf{v}_{<i}))}$$

for $i \in \{1, ...D\}$, where $\mathbf{v}_{<i}$ is the subvector consisting of all $v_q$ such that $q < i$ i.e., $\mathbf{v}_{<i} \in \{v_1, ..., v_{i-1}\}$, $g(\cdot)$ is a non-linear activation function, $\mathbf{W} \in \mathbb{R}^{H \times K}$ and $\mathbf{U} \in \mathbb{R}^{K \times H}$ are weight matrices, $\mathbf{c} \in \mathbb{R}^H$ and $\mathbf{b} \in \mathbb{R}^K$ are bias parameter vectors. $H$ is the number of hidden units (topics).

Figure 1 (left) (without *WordPool*) provides an illustration of the $i$th autoregressive step of the Doc-NADE architecture, where the parameter $\mathbf{W}$ is shared in the feed-forward networks and $\mathbf{h}_i$ encodes topic-proportion embedding. Importantly, the topic-word matrix $\mathbf{W}$ has a property that the column vector $\mathbf{W}_{:,v_i}$ corresponds to embedding of the word $v_i$, whereas the row vector $\mathbf{W}_{j,:}$ encodes the $j$th topic. We leverage this property to introduce external knowledge via word and topic embeddings.

Additionally, DocNADE has shown to outperform traditional models such as LDA (Blei et al., 2003) and RSM (Salakhutdinov & Hinton, 2009) in terms of both the log-probability on unseen documents and retrieval accuracy. Recently, Gupta et al. (2019) has improved topic modeling on short texts by introducing word embeddings (Pennington et al., 2014) in DocNADE architecture. Thus, we adopt DocNADE to perform transfer learning within the neural topic modeling framework.

Algorithm 1 (for DocNADE, set `LVT` and `GVT` to *False*) demonstrates the computation of $\log p(\mathbf{v})$ and negative log-likelihood $\mathcal{L}(\mathbf{v})$ that is minimized using gradient descent. Moreover, computing $\mathbf{h}_i$ is efficient (linear complexity) due to NADE architecture that leverages the pre-activation $\mathbf{a}_{i-1}$ of $(i-1)$th step in computing the pre-activation $\mathbf{a}_i$. See Larochelle & Lauly (2012) for further details.

## 2.2 Multi-View (MVT) and Multi-Source Transfers (MST) in Topic Modeling

Here, we describe a transfer learning framework in topic modeling that jointly exploits the complementary knowledge using the *WordPool* and *TopicPool*, the KBs of pretrained word and (latent) topic embeddings, respectively obtained from large document collections (DCs) from several sources. In

| Target Domain Corpora | | | | | | | | | Source Domain Corpora | | | | | | | |
|---|---|---|---|---|---|---|---|---|---|---|---|---|---|---|---|---|
| **ID** | **Data** | **Train** | **Val** | **Test** | $K$ | **L** | **C** | | **ID** | **Data** | **Train** | **Val** | **Test** | $K$ | **L** | **C** |
| $\mathcal{T}^1$ | 20NSshort | 1.3k | 0.1k | 0.5k | 1.4k | 13.5 | 20 | | $\mathcal{S}^1$ | 20NS | 7.9k | 1.6k | 5.2k | 2k | 107.5 | 20 |
| $\mathcal{T}^2$ | 20NSsmall | 0.4k | 0.2k | 0.2k | 2k | 187.5 | 20 | | $\mathcal{S}^2$ | R21578 | 7.3k | 0.5k | 3.0k | 2k | 128 | 90 |
| $\mathcal{T}^3$ | TMNtitle | 22.8k | 2.0k | 7.8k | 2k | 4.9 | 7 | | $\mathcal{S}^3$ | TMN | 22.8k | 2.0k | 7.8k | 2k | 19 | 7 |
| $\mathcal{T}^4$ | R21578title | 7.3k | 0.5k | 3.0k | 2k | 7.3 | 90 | | $\mathcal{S}^4$ | AGNews | 118k | 2.0k | 7.6k | 5k | 38 | 4 |
| $\mathcal{T}^5$ | Ohsumedtitle | 8.3k | 2.1k | 12.7k | 2k | 11.9 | 23 | | $\mathcal{S}^5$ | PubMed | 15.0k | 2.5k | 2.5k | 3k | 254.8 | - |
| $\mathcal{T}^6$ | Ohsumed | 8.3k | 2.1k | 12.7k | 3k | 159.1 | 23 | | | | | | | | | |

|  | $\mathcal{T}^1$ | $\mathcal{T}^2$ | $\mathcal{T}^3$ | $\mathcal{T}^4$ | $\mathcal{T}^5$ | $\mathcal{T}^6$ |
|---|---|---|---|---|---|---|
| $\mathcal{S}^1$ | $\mathcal{I}$ | $\mathcal{I}$ | $\mathcal{R}$ | $\mathcal{D}$ | $\mathcal{D}$ | $\mathcal{D}$ |
| $\mathcal{S}^2$ | $\mathcal{D}$ | $\mathcal{D}$ | $\mathcal{D}$ | $\mathcal{I}$ | $\mathcal{D}$ | $\mathcal{D}$ |
| $\mathcal{S}^3$ | $\mathcal{R}$ | $\mathcal{R}$ | $\mathcal{I}$ | $\mathcal{D}$ | $\mathcal{D}$ | $\mathcal{D}$ |
| $\mathcal{S}^4$ | $\mathcal{R}$ | $\mathcal{R}$ | $\mathcal{R}$ | $\mathcal{D}$ | $\mathcal{D}$ | $\mathcal{D}$ |
| $\mathcal{S}^5$ | $\mathcal{D}$ | $\mathcal{D}$ | $\mathcal{D}$ | $\mathcal{D}$ | - | - |

Table 2: Data statistics: Short/long texts and/or small/large corpora in target and source domains. Symbols- $K$: vocabulary size, $L$: average text length (#words), $C$: number of classes and $k$: thousand. For short-text, $L<15$. $\mathcal{S}^3$ is also used in target domain. '-': unlabeled data.

Table 3: Domain overlap in source-target corpora. $\mathcal{I}$: Identical, $\mathcal{R}$: Related and $\mathcal{D}$: Distant domains.

doing so, we first apply the DocNADE to generate a topic-word matrix for each of the DCs, where its column-vector and row-vector generate $\mathbf{E}^k$ and $\mathbf{Z}^k$, respectively for the $k$th source.

**LVT+MST Formulation for Multi-source Word Embedding Transfer**: As illustrated in Figure 1 (left) and Algorithm 1 with LVT=*True*, we perform transfer learning on a target $\mathcal{T}$ using the *WordPool* of pretrained word embeddings $\{\mathbf{E}^1, ..., \mathbf{E}^{|\mathcal{S}|}\}$ from several sources $\mathcal{S}$ (i.e., multi-source):

$$\mathbf{h}_i(\mathbf{v}_{<i}) = g(\mathbf{c} + \sum_{q<i} \mathbf{W}_{:,v_q} + \sum_{q<i} \sum_{k=1}^{|\mathcal{S}|} \lambda^k \, \mathbf{E}^k_{:,v_q})$$

Here, $k$ refers to the $k$th source and $\lambda^k$ is a weight for $\mathbf{E}^k$ that controls the amount of knowledge transferred in $\mathcal{T}$, based on domain overlap between target and source(s). Recently, DocNADEe (Gupta et al., 2019) has incorporated word embeddings (Pennington et al., 2014) in extending Doc-NADE; however, it is based on a *single* source.

**GVT+MST Formulation for Multi-source Topic Embedding Transfer**: Next, we perform knowledge transfer exclusively using the *TopicPool* of pretrained topic embeddings (e.g., $\mathbf{Z}^k$) from one or several sources, $\mathcal{S}$. In doing so, we add a regularization term to the loss function $\mathcal{L}(\mathbf{v})$ and require DocNADE to minimize the overall loss in a way that the (latent) topic features in $\mathbf{W}$ simultaneously inherit relevant topical features from each of the source domains $\mathcal{S}$, and generate meaningful representations for the target $\mathcal{T}$. The overall loss $\mathcal{L}(\mathbf{v})$ due to GVT+MST in DocNADE is given by:

$$\mathcal{L}(\mathbf{v}) = -\log p(\mathbf{v}) + \sum_{k=1}^{|\mathcal{S}|} \gamma^k \sum_{j=1}^{H} ||\mathbf{A}^k_{j,:}\mathbf{W} - \mathbf{Z}^k_{j,:}||_2^2$$

Here, $\mathbf{A}^k \in \mathbb{R}^{H \times H}$ aligns latent topics in the target $\mathcal{T}$ and $k$th source, and $\gamma^k$ governs the degree of imitation of topic features $\mathbf{Z}^k$ by $\mathbf{W}$ in $\mathcal{T}$. Consequently, the generative process of learning meaningful topics in $\mathbf{W}$ of $\mathcal{T}$ is guided by relevant features in $\{\mathbf{Z}\}_1^{|\mathcal{S}|}$ to address data-sparsity. Algorithm 1 describes the computation of the loss, when GVT = *True* and LVT = *False*.

Moreover, Figure 1 (right) illustrates the need for topic alignments between target and source(s). Here, $j$ indicates the topic (i.e., row) index in a topic matrix, e.g., $\mathbf{Z}^k$. Observe that the first topic (gray curve), i.e., $Z^1_{j=1} \in \mathbf{Z}^1$ of the first source aligns with the first row-vector (i.e., topic) of $\mathbf{W}$ (of target). However, the other two topics $Z^1_{j=2}, Z^1_{j=3} \in \mathbf{Z}^1$ need alignment with the target topics.

**MVT+MST Formulation for Multi-source Word and Topic Embeddings Transfer**: When LVT and GVT are *True* (Algorithm 1) for many sources, the two complementary representations are jointly used in transfer learning using *WordPool* and *TopicPool*, and therefore, the name *multi-view* and *multi-source* transfers.

## 3 EVALUATION AND ANALYSIS

**Datasets**: Table 2 describes the datasets used in high-resource source and low-and high-resource target domains for our experiments. The target domain $\mathcal{T}$ consists of four short-text corpora (20NSshort, TMNtitle, R21578title and Ohsumedtitle), one small corpus (20NSsmall) and

| Baselines (Related Works) | Features | | | |
|---|---|---|---|---|
| | *NTM* | *AuR* | *LVT* | *GVT*\|*MVT*\|*MST* |
| LDA (Blei et al., 2003) | | | | |
| RSM (Salakhutdinov & Hinton, 2009) | ✓ | | | |
| DocNADE (Larochelle & Lauly, 2012) | ✓ | ✓ | | |
| NVDM (Miao et al., 2016) | ✓ | | | |
| ProdLDA (Srivastava & Sutton, 2017) | | | | |
| Gauss-LDA (Das et al., 2015) | | | ✓ | |
| glove-DMM (Nguyen et al., 2015) | | | ✓ | |
| DocNADEe (Gupta et al., 2019) | ✓ | ✓ | ✓ | |
| EmbSum | | | | |
| doc2vec (Le & Mikolov, 2014) | | | | |
| **this work** | ✓ | ✓ | ✓ | ✓ ✓ ✓ |

Table 4: Baselines (related works) vs this work. Here, *NTM* and *AuR* refer to neural network-based TM and autoregressive assumption, respectively. DocNADEe → DocNADE+Glove embeddings.

two large corpora (`TMN` and `Ohsumed`). However in source $\mathcal{S}$, we use five large corpora (`20NS`, `R21578`, `TMN`, `AGnews` and `PubMed`) in different label spaces (i.e, domains). Here, the corpora ($\mathcal{T}^5$, $\mathcal{T}^6$ and $\mathcal{S}^5$) belong to *medical* and others to *news*.

Additionally, Table 3 suggests domain overlap (in terms of label match) in the target and source corpora, where we define three types of overlap: $\mathcal{I}$ (identical) if all labels match, $\mathcal{R}$ (related) if some labels match, and $\mathcal{D}$ (distant) if a very few or no labels match. Note, our modeling approaches are completely unsupervised and do not use the data labels. See the data labels in *appendices*.

**Reproducibility**: For evaluations in the following sections, we follow the experimental setup similar to DocNADE (Larochelle & Lauly, 2012) and DocNADEe (Gupta et al., 2019), where the number of topics ($H$) is set to 200. While DocNADEe requires the dimension (i.e., $E$) of word embeddings be the same as the latent topic (i.e., $H$), we first apply a projection on the concatenation of the pre-trained word embeddings obtained from several sources and then, introduce the prior knowledge in each of the autoregressive step following DocNADEe. We apply it in configurations where Glove and/or FastText ($E$=300) (Bojanowski et al., 2017) are employed. See *appendices* for the experimental setup, hyperparameters[1] and optimal values of $\lambda^k \in [0.1, 0.5, 1.0]$ and $\gamma^k \in [0.1, 0.01, 0.001]$ (determined using development set) in different source-target configurations. (code provided)

**Baselines**: As summarized in Table 4, we consider several baselines including (1) LDA-based and neural network-based topic models that use the target data, (2) topic models using pretrained word embeddings (i.e., LVT) from Pennington et al. (2014) (Glove), (3) unsupervised document representation, where we employ doc2vec (Le & Mikolov, 2014) and EmbSum (to represent a document by summing the embedding vectors of its words using Glove) in order to quantify the quality of document representations, (4) zero-shot topic modeling, where we use all source corpora and no target corpus, and (5) data-augmentation, where we use all source corpora along with a target corpus for TM on $\mathcal{T}$. Using DocNADE, we first prepare the two KBs: *WordPool* and *TopicPool* from each of the source corpora and then use them in knowledge transfer to $\mathcal{T}$.

Tables 5 and 6 show the comparison of our proposed transfer learning approaches (i.e., LVT using *WordPool*, GVT using *TopicPool*, MVT and MST) with the baselines TMs that (1) do not, and (2) do employ pretrained word embeddings (e.g., DocNADE and DocNADEe, respectively).

## 3.1 Generalization: Perplexity (PPL)

To evaluate generative performance of TM, we estimate the log-probabilities for the test documents and compute the average held-out perplexity per word as, $PPL = \exp\left(-\frac{1}{N}\sum_{t=1}^{N}\frac{1}{|\mathbf{v}_t|}\log p(\mathbf{v}_t)\right)$, where $N$ and $|\mathbf{v}_t|$ are the number of documents and words in a document $\mathbf{v}_t$, respectively.

---

[1]selected with grid search; suboptimal results (see *appendices*) by learning $\lambda$ and $\gamma$ with backpropagation

| KBs from Source Corpus | Model/ Transfer Type | Scores on Target Corpus (*in sparse-data and sufficient-data settings*) | | | | | | | | | | | | | |
|---|---|---|---|---|---|---|---|---|---|---|---|---|---|---|---|
| | | 20NSshort | | | TMNtitle | | | R21578title | | | 20NSsmall | | | TMN | |
| | | *PPL* | *COH* | *IR* | *PPL* | *COH* | *IR* | *PPL* | *COH* | *IR* | *PPL* | *COH* | *IR* | *PPL* | *COH* |
| *Baseline TM* | NVDM | 1047 | .736 | .076 | 973 | .740 | .190 | 372 | .735 | .271 | 957 | .515 | .090 | 833 | .673 |
| **without** *Word-* | ProdLDA | 923 | .689 | .062 | 1527 | .744 | .170 | 480 | .742 | .200 | 1181 | .394 | .062 | 1519 | .577 |
| *Embeddings* | DocNADE | 646 | .667 | .290 | 706 | .709 | .521 | 192 | .713 | .657 | 594 | .462 | .270 | 584 | .636 |
| | LVT | 630 | .673 | .298 | 705 | .709 | .523 | 194 | .708 | .656 | 594 | .455 | .288 | 582 | .649 |
| *20NS* | GVT | 646 | .690 | .303 | 718 | .720 | .527 | 184 | .698 | .660 | 594 | .500 | .310 | 590 | .652 |
| | MVT | **638** | .690 | **.314** | 714 | .718 | .528 | 188 | .715 | .655 | 600 | .499 | .311 | 588 | .650 |
| | LVT | 649 | .668 | .296 | **655** | .731 | .548 | 187 | .703 | .659 | 593 | .460 | .273 | - | - |
| *TMN* | GVT | 661 | .692 | .294 | 689 | .728 | .555 | 191 | .709 | .660 | 596 | .521 | .276 | - | - |
| | MVT | 658 | .687 | .297 | 663 | .747 | .553 | 195 | .720 | .660 | 599 | .507 | .292 | - | - |
| | LVT | 656 | .667 | .292 | 704 | .715 | .522 | 186 | .715 | .676 | 593 | .458 | .267 | 581 | .636 |
| *R21578* | GVT | 654 | .672 | .293 | 716 | .719 | .526 | 194 | .706 | .672 | 595 | .485 | .279 | 591 | .646 |
| | MVT | 650 | .670 | .296 | 716 | .720 | .528 | 194 | .724 | .676 | 599 | .490 | .280 | 589 | .650 |
| | LVT | 650 | .677 | .297 | 682 | .723 | .533 | 185 | .710 | .659 | **592** | .458 | .260 | **564** | .668 |
| *AGnews* | GVT | 667 | .695 | .300 | 728 | .735 | .534 | 190 | .717 | .663 | 598 | .563 | .282 | 601 | .684 |
| | MVT | 659 | .696 | .290 | 718 | .740 | .533 | 189 | .727 | .659 | 599 | .566 | .279 | 592 | .686 |
| | LVT | 640 | .678 | .308 | 663 | .732 | .547 | **182** | .739 | .673 | 594 | .542 | .277 | 568 | .674 |
| *MST* | GVT | 658 | .705 | .305 | 704 | .746 | .550 | 192 | .727 | .673 | 599 | .585 | **.326** | 602 | .680 |
| | MVT | 656 | **.740** | **.314** | 680 | **.752** | **.569** | 188 | **.745** | **.685** | 600 | **.637** | .285 | 600 | **.690** |
| **Gain%** (vs DocNADE) | | 1.23 | 10.9 | 8.28 | 7.22 | 6.06 | 9.21 | 5.20 | 4.49 | 4.26 | 0.34 | 37.9 | 20.7 | 3.42 | 8.50 |

Table 5: State-of-the-art comparisons with TMs: Perplexity (PPL), topic coherence (COH) and precision (IR) at retrieval fraction 0.02. Scores are reported on each of the target, given KBs from one or several sources. *Please read column-wise.* **Bold**: best in column. Gain%: Bold vs DocNADE.

| KBs from Source Corpus | Model/ Transfer Type | Scores on Target Corpus (*in sparse-data and sufficient-data settings*) | | | | | | | | | | | | | |
|---|---|---|---|---|---|---|---|---|---|---|---|---|---|---|---|
| | | 20NSshort | | | TMNtitle | | | R21578title | | | 20NSsmall | | | TMN | |
| | | *PPL* | *COH* | *IR* | *PPL* | *COH* | *IR* | *PPL* | *COH* | *IR* | *PPL* | *COH* | *IR* | *PPL* | *COH* |
| | doc2vec | - | - | .090 | - | - | .190 | - | - | .518 | - | - | .200 | - | - |
| | EmbSum | - | - | .236 | - | - | .513 | - | - | .587 | - | - | .214 | - | - |
| *Baseline TM* | Gauss-LDA | - | - | .080 | - | - | .408 | - | - | .367 | - | - | .090 | - | - |
| **with** *Word-* | glove-DMM | - | .512 | .183 | - | .633 | .445 | - | .364 | .273 | - | .578 | .090 | - | .705 |
| *Embeddings* | DocNADEe | **629** | .674 | .294 | 680 | .719 | .540 | 187 | .721 | .663 | **590** | .455 | .274 | 572 | .664 |
| *20NS* | MVT+Glove | 630 | .721 | **.320** | 688 | .741 | .565 | **183** | .724 | .667 | 597 | .561 | .306 | **570** | .693 |
| *TMN* | MVT+Glove | 640 | .731 | .295 | **673** | .750 | .576 | 184 | .716 | .672 | 599 | .594 | .261 | - | - |
| *R21578* | MVT+Glove | 633 | .705 | .295 | 689 | .738 | .540 | 185 | .737 | **.691** | 595 | .485 | .255 | 577 | .697 |
| *AGnews* | MVT+Glove | 642 | .734 | .302 | 706 | .748 | .565 | 190 | .734 | .675 | 598 | .573 | .284 | 585 | .703 |
| *MST* | MVT+Glove | 644 | .739 | .304 | **673** | **.752** | .570 | **183** | .742 | .684 | 598 | .631 | **.282** | 582 | .710 |
| | + FastText | 654 | **.741** | .313 | **673** | .751 | **.578** | **183** | **.744** | .684 | 599 | **.634** | .254 | 582 | **.711** |
| **Gain%** (vs DocNADEe) | | - | 9.95 | 8.84 | 1.03 | 4.60 | 7.04 | 2.14 | 3.20 | 4.22 | - | 39.3 | 2.92 | .35 | 7.08 |

Table 6: State-of-the-art comparisons with TMs using word embeddings: PPL, COH and IR at retrieval fraction 0.02. Scores are reported on each of the target, given KBs. Here, MVT: LVT+GVT (Table 5), DocNADEe: DocNADE+Glove and Gain%: Bold vs DocNADEe. For all the configurations, we apply a projection on word embeddings concatenated from several sources.

Tables 5 and 6 quantitatively show PPL scores on the five target corpora (four short-text and one long-text) by the baselines and proposed transfer learning approaches (i.e., GVT, MVT and MST) using one or four sources. In Table 5 using TMN (as a single source) for LVT, GVT and MVT on TMNtitle, we see improved (reduced) PPL scores: (655 vs 706), (689 vs 706) and (663 vs 706) respectively in comparison to DocNADE. We also observe gains due to MST+LVT, MST+GVT and MST+MVT configurations on TMNtitle. Similarly in MST+LVT for R21578title, we observe a gain of 5.2% (182 vs 192), suggesting that transfer learning using pretrained word and topic embeddings (jointly) from one or many sources helps due to positive knowledge transfer, and it also verifies domain relatedness (e.g., in TMN-TMNtitle and AGnews-TMN). Similarly, Table 6 shows gains in PPL (e.g., on TMNtitle, R21578title, etc.) compared to DocNADEe.

| KBs from Source Corpus | Model/ Transfer Type | Scores on Target Corpus | | | | | |
|---|---|---|---|---|---|---|---|
| | | Ohsumedtitle | | | Ohsumed | | |
| | | PPL | COH | IR | PPL | COH | IR |
| baselines | ProdLDA | 1121 | .734 | .080 | 1677 | .646 | .080 |
| | DocNADE | 1321 | .728 | .160 | 1706 | .662 | .184 |
| | EmbSum | - | - | .150 | - | - | .148 |
| | DocNADEe | 1534 | .738 | .175 | 1637 | .674 | .183 |
| AGnews | LVT | 1587 | .732 | .160 | 1717 | .657 | .184 |
| | GVT | 1529 | .732 | .160 | 1594 | .665 | .185 |
| | MVT | 1528 | .734 | .160 | 1598 | .666 | .184 |
| | + BioEmb | 1488 | .747 | .176 | 1595 | .681 | .187 |
| PubMed | LVT | **1268** | .732 | .172 | 1535 | .669 | .190 |
| | GVT | 1392 | .740 | .173 | 1718 | .671 | **.192** |
| | MVT | 1408 | .743 | .178 | 1514 | .674 | .191 |
| | + BioEmb | 1364 | **.753** | **.182** | 1633 | **.689** | .191 |
| MST | LVT | **1268** | .733 | .172 | 1536 | .668 | .190 |
| | GVT | 1391 | .740 | .172 | 1504 | .666 | **.192** |
| | MVT | 1399 | .744 | .177 | 1607 | .679 | .191 |
| | + BioEmb | 1375 | .751 | .180 | **1497** | .693 | .190 |
| | + BioFastText | 1350 | **.753** | .178 | 1641 | .688 | .187 |
| **Gain%** (vs DocNADE) | | 4.01 | 3.43 | 13.8 | 12.3 | 4.08 | 4.35 |
| **Gain%** (vs DocNADEe) | | 17.3 | 2.03 | 4.00 | 8.55 | 2.22 | 4.91 |

Table 7: PPL, COH, IR at retrieval fraction 0.02. BioEmb and BioFastText: 200-dimensional word vectors from large biomedical corpus (Moen & Ananiadou, 2013). + BioEmb: MVT+BioEmb.

| $\mathcal{T}$ | $\mathcal{S}$ | Model | Topic-words (Top 5) |
|---|---|---|---|
| 20NSshort | 20NS | DNE | shipping, sale, prices, expensive, price |
| | | -GVT | sale, price, monitor, site, setup |
| | | +GVT | shipping, sale, price, expensive, subscribe |
| | AGnews | DNE | microsoft, software, ibm, linux, computer |
| | | -GVT | apple, modem, side, baud, perform |
| | | +GVT | microsoft, software, desktop, computer, apple |
| TMNtitle | AGnews | DNE | miners, earthquake, explosion, stormed, quake |
| | TMN | DNE | tsunami, quake, japan, earthquake, radiation |
| | | -GVT | strike, jackson, kill, earthquake, injures |
| | | +GVT | earthquake, radiation, explosion, wildfire |

Table 8: Source $\mathcal{S}$ and target $\mathcal{T}$ topics before (-) and after (+) topic transfer(s) (GVT) from one or more sources. DNE: DocNADE

| chip | | | | |
|---|---|---|---|---|
| source corpora | | | target corpus | |
| | | | 20NSshort | |
| 20NS | R21578 | AGnews | -GVT | +GVT |
| key | chips | chips | virus | chips |
| encrypted | semiconductor | chipmaker | intel | technology |
| encryption | miti | processors | gosh | intel |
| clipper | makers | semiconductor | crash | encryption |
| keys | semiconductors | intel | chips | clipper |

Table 9: Five nearest neighbors of the word *chip* in source and target semantic spaces before (-) and after (+) knowledge transfer (MST+GVT)

In Table 7, we show PPL scores on two medical target corpora: Ohsumtitle and Ohsumed using two sources: AGnews (*news corpus*) and PubMed (*medical* abstracts) to perform cross-domain and in-domain knowledge transfers. We see that using PubMed for LVT on both the target corpora improves generalization. Overall, we report a gain of 17.3% (1268 vs 1534) on Ohsumtitle and 8.55% (1497 vs 1637) on Ohsumtitle, compared to DocNADEe. Additionally, MST+GVT and MST+MVT boost generalization performance compared to DocNADE(e).

## 3.2 INTERPRETABILTY: TOPIC COHERENCE (COH)

While PPL is used for model selection, adjusting parameters (e.g. $H$) and quantitative comparisons, Chang et al. (2009) showed in some cases humans preferred TMs (based on the semantic quality of topics) with higher (worse) PPLs. Thus beyond perplexity, we compute topic coherence to estimate the meaningfulness of words in each of the topics captured. In doing so, we choose the coherence measure proposed by Röder et al. (2015) that identifies context features for each topic word using a sliding window over the reference corpus. We follow Gupta et al. (2019) and compute COH with the top 10 words in each topic. Essentially, the higher scores imply the more coherent topics.

Tables 5 and 6 (under COH column) demonstrate that our proposed approaches (GVT, MVT and MST) of transfer learning in TMs show noticeable gains in COH and thus, improve topic quality. For instance in Table 5, when AGnews is used as a single source for 20NSsmall datatset, we observe a gain in COH due to GVT (.563 vs .462) and MVT (.566 vs .462). Additionally, noticeable gains are reported due to MST+LVT (.542 vs .462), MST+GVT (.585 vs .462) and MST+MVT (.637 vs .462), compared to DocNADE. Importantly, we find a trend MVT>GVT>LVT in COH scores for both the single-source and multi-source transfers. Similarly, Table 6 show noticeable gains (e.g., 39.3%, 9.95%, 7.08%, etc.) in COH due to MST and MVT with Glove and FastText word embeddings. Moreover, Table 7 shows gains in COH due to GVT on Ohsumedtitle and Ohsumed, using pretrained knowledge from PubMed. Overall, the GVT, MVT and MST boost COH for all the five target corpora compared to the baseline TMs (i.e., DocNADE and DocNADEe). This suggests that there is a need for the two complementary (pretrained word and topics) representations and multi-source transfer learning in order to guide meaningful topic learning in $\mathcal{T}$. The results on both the low- and high-resource targets across domains conclude that the proposed modeling scales.

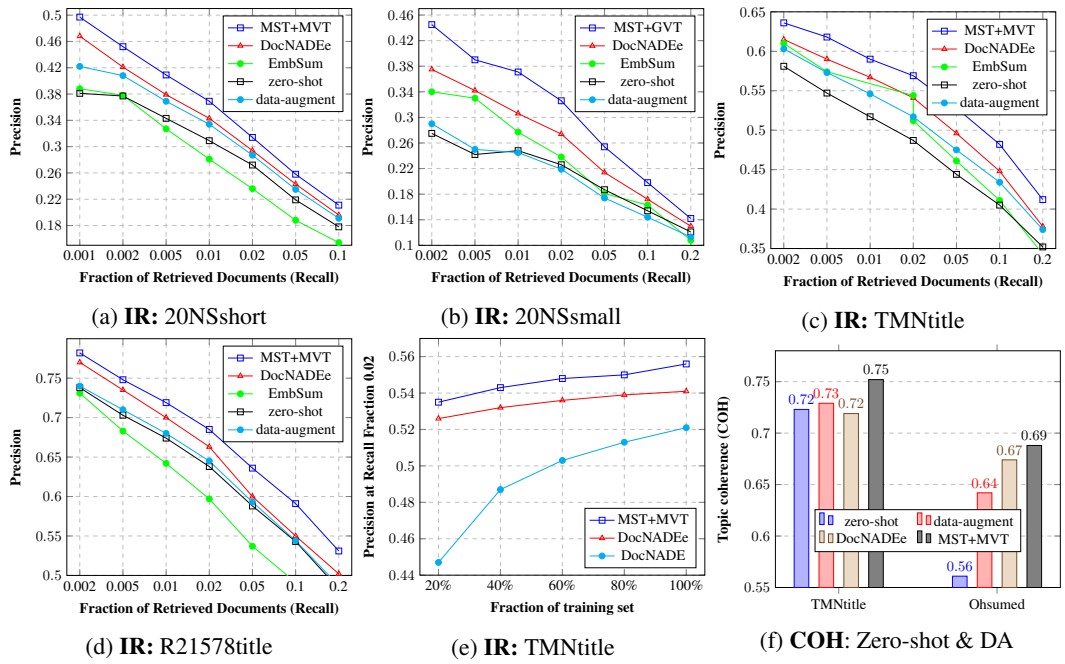

Figure 2: (a, b, c, d) Retrieval performance (precision) on four datasets. (e) Precision at recall fraction 0.02, each for a fraction (20%, 40%, 60%, 80%, 100%) of the training set of TMNtitle. (f) Zero-shot and data-augmentation (DA) experiments for topic coherence on TMNtitle and Ohsumed.

## 3.3 APPLICABILITY: INFORMATION RETRIEVAL (IR)

For a greater impact of TMs, we further evaluate the quality of document representations and perform a document retrieval task on the target datasets, using their label information only to compute precision. We follow the experimental setup similar to Lauly et al. (2017), where all test documents are treated as queries to retrieve a fraction of the closest documents in the original training set using cosine similarity between their document vectors. To compute retrieval precision for each fraction (e.g., 0.02), we average the number of retrieved training documents with the same label as the query.

Tables 5 and 6 depict precision scores at retrieval fraction 0.02 (similar to Gupta et al. (2019)), where the configuration MST+MVT outperforms both the DocNADE and DocNADEe, respectively in retrieval performance on the four target (short-text) datasets. A gain in IR performance is noticeable for highly overlapping domains, e.g., TMN-TMNtitle (.555 vs .521 in Table 5 and .576 vs .540 in Table 6) than the related, e.g., AGnews-TMNtitle (.534 vs .521 in Table 5 and .565 vs .540 in Table 6). We observe large gains in precision at retrieval fraction 0.02: (a) Table 5: 20.7% (.326 vs .270) on 20NSsmall, 9.21% (.569 vs .521) on TMNtitle and 8.28% (.314 vs .290) on 20NSshort, (b) Table 6: 8.84% (.320 vs .294) on 20NSshort and 9.21% (.578 vs .540) on TMNtitle, and (c) Table 7: 4.91% (.192 vs .183) on Ohsumed and 4.0% (.182 vs .175) on Ohsumedtitle.

Additionally, Figures 2a, 2b, 2c and 2d illustrate precision on 20NSshort, 20NSsmall, TMNtitle and R21578title, respectively, where our approaches (MST+GVT and MST+MVT) consistently outperform the baselines at all fractions. Moreover, we split the training data of TMNtitle into several sets: 20%, 40%, 60%, 80% of the training set and then retrain DocNADE, DocNADEe and DocNADE+MST+MVT. We demonstrate the impact of transfer learning in sparse-data settings using *WordPool* and *TopicPool* jointly on IR task. Figure 2e plots precision at retrieval (recall) fraction 0.02 and demonstrates that the proposed modeling consistently outperform DocNADE(e).

## 3.4 ZERO-SHOT AND DATA-AUGMENTATION EVALUATIONS

Figures 2a, 2b, 2c and 2d show precision in the zero-shot (source-only training) and data-augmentation (source+target training) configurations. Observe that the latter helps in learning

meaningful representations and performs better than zero-shot; however, it is outperformed by MST+MVT, suggesting that a naive (data space) augmentation does not add sufficient prior or relevant information to the sparse target. Thus, we find that it is beneficial to augment training data in feature space (e.g., LVT, GVT and MVT) especially for unsupervised TMs using *WordPool* and *TopicPool*. Beyond IR, we further investigate computing topic coherence (COH) for zero-shot and data-augmentation baselines, where the COH scores (Figure 2f) suggest that MST+MVT outperforms DocNADEe, zero-shot and data-augmentation.

### 3.5 QUALITATIVE ANALYSIS: TOPICS AND NEAREST NEIGHBORS (NN)

For topic level inspection, we first extract topics using the rows of $\mathbf{W}$ of source and target corpora. Table 8 shows the topics (top-5 words) from source and target domains. Observe that the target topics become more coherent after transfer learning (i.e., +GVT) from one or more sources. The blue color signifies that a target topic has imitated certain topic words from the source. Observe that we also show topics from source domain(s) that align with the topics from target.

For word level inspection, we extract word representations using the columns of $\mathbf{W}$. Table 9 shows nearest neighbors (NNs) of the word *chip* in `20NSshort` (target) corpus, before and after GVT using three knowledge sources. Observe that the NNs in the target become more meaningful.

## 4 CONCLUSION

Within neural topic modeling, we have introduced transfer learning approaches using complementary representations: pretrained word (local semantics) and topic (global semantics) embeddings exclusively or jointly from one or many sources (i.e., multi-view and multi-source). We have shown that the proposed approaches better deal with data-sparsity issues, especially in a short-text and/or small document collection. We have demonstrated learning meaningful topics and quality document representations on 7 (low- and high-resource) target corpora from news and medical domains.

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

## A  DATA DESCRIPTION

In order to evaluate knowledge transfer within unsupervised neural topic modeling, we use the following seven datasets in the target domain $\mathcal{T}$ following the similar experimental setup as in DocNADEe:

1. 20NSshort: We take documents from 20NewsGroups data, with document size less (in terms of number of words) than 20.

2. 20NSsmall: We sample 20 document (each having more than 200 words) for training from each class of the 20NS dataset. For validation and test, 10 document for each class. Therefore, it is a corpus of few (long) documents.

3. TMNtitle: Titles of the Tag My News (TMN) news dataset.

4. R21578title: Reuters corpus, a collection of new stories from nltk.corpus. We take titles of the documents.

5. Ohsumedtitle: Titles of Ohsumed abstracts. Source: disi.unitn.it/moschitti/corpora.htm.

6. Ohsumed: Ohsumed dataset, collection of medical abstracts. Source: disi.unitn.it/moschitti/corpora.htm.

7. TMN: The Tag My News (TMN) news dataset.

To prepare knowledge base of word embedings (local semantics) and latent topics (global semantics) features, we use the following six datasets in the source $\mathcal{S}$:

1. 20NS: 20NewsGroups corpus, a collection of news stories from nltk.corpus.

2. TMN: The Tag My News (TMN) news dataset.

3. R21578: Reuters corpus, a collection of new stories from nltk.corpus.

4. AGnews: AGnews data sellection.

5. PubMed: Medical abstracts of randomized controlled trials. Source: https://github.com/Franck-Dernoncourt/pubmed-rct.

## B  GETTING WORD AND LATENT TOPIC REPRESENTATIONS FROM SOURCE(S)

Since in DocNADE, the column of $\mathbf{W}_{:,v_i}$ gives a word vector of the word $v_i$, therefore the dimension of word embeddings in each of the $\mathbf{E}^k$ is same (i.e., $H = 200$). Thus, we prepare the knowledge base of word representations $\mathbf{E}^k$ from $k$th source using DocNADE, where each word vector is of $H = 200$ dimension.

Since the row vector of $\mathbf{W}_{j,:}$ in DocNADE encodes $j$th topic feature, therefore each latent topic (i.e., row) in feature matrix $\mathbf{W}$ is a vector of $K$ dimension, corresponding the definition of topics that it is a distribution over vocabulary. $H$ is the number of latent topics and $K$ is the vocabulary size, where $K$ varies across corpora. Thus, we train DocNADE to learn a feature matrix specific to each of the source corpora, e.g. $\mathbf{W}^k \in \mathbb{R}^{H \times K}$ of $k$th source.

For a target corpus of vocabulary size $K^{'}$, the DocNADE learns a feature matrix $\mathbf{W}^{\mathcal{T}} \in \mathbb{R}^{H \times K^{'}}$. Similarly, $\mathbf{W}^k \in \mathbb{R}^{H \times K}$ for $k$th source of vocabulary size $K$. Since in the sparse-data setting for the

| data | labels / classes |
|------|------------------|
| TMNtitle and TMN | world, us, sport, business, sci_tech, entertainment, health |
| AGnews | business, sci_tech, sports, world |
| 20NSshort, 20NSsmall, 20NS | misc.forsale, comp.graphics, rec.autos, comp.windows.x, rec.sport.baseball, sci.space, rec.sport.hockey, soc.religion.christian, rec.motorcycles, comp.sys.mac.hardware, talk.religion.misc, sci.electronics, comp.os.ms-windows.misc, sci.med, comp.sys.ibm.pc.hardware, talk.politics.mideast, talk.politics.guns, talk.politics.misc, alt.atheism, sci.crypt |
| R21578title and R21578 | trade, grain, crude, nat-gas, corn, rice, rubber, sugar, tin, palm-oil, veg-oil, ship, coffee, lumber, wheat, gold, acq, interest, money-fx, copper, ipi, carcass, livestock, oilseed, soybean, earn, bop, gas, lead, jobs, zinc, cpi, gnp, soy-oil, dlr, yen, nickel, groundnut, heat, sorghum, sunseed, pet-chem, cocoa, rapeseed, cotton, money-supply, iron-steel, l-cattle, alum, palladium, platinum, strategic-metal, reserves, groundnut-oil, lin-oil, meal-feed, rape-oil, sun-meal, sun-oil, hog, barley, potato, orange, retail, soy-meal, cotton-oil, oat, fuel, silver, income, wpi, tea, lei, coconut, coconut-oil, copra-cake, dfl, dmk, naphtha, propane, instal-debt, nzdlr, housing, nkr, rye, castor-oil, jet, palmkernel, cpu, rand |

Table 10: Label space of the corpora used

| Hyperparameter | Search Space |
|----------------|--------------|
| retrieval fraction | [0.02] |
| learning rate | [0.001] |
| hidden units, $H$ | [200] |
| activation function ($g$) | sigmoid |
| iterations | [100] |
| $\lambda^k$ | [1.0, 0.5, 0.1] |
| $\gamma^k$ | [0.1, 0.01, 0.001] |

Table 11: Hyperparameters in Generalization in DocNADE, DocNADEe, LVT, GVT and MVT configurations for 200 topics

target, $K' << K$ due to additional word in the source. In order to perform GVT, we need the same topic feature dimensions in the target and source, i.e., $K'$ of the target. Therefore, we remove those column vectors from $\mathbf{W}^k \in \mathbb{R}^{H \times K}$ of the $k$th source for which there is no corresponding word in the vocabulary of the target domain. As a result, we obtain $\mathbf{Z}^k$ as a latent topic feature matrix to be used in knowledge transfer to the target domain. Following the similar steps, we prepare a KB of $\mathbf{Z}$s such that each latent topic feature matrix from a source domain gets the same topic feature dimension as the target.

## C  EXPERIMENTAL SETUP

For DocNADE and DocNADEe in different knowledge transfer configurations, we follow the same experimental setup as in DocNADE and DocNADEe. We rerun DocNADE and DocNADEe using the code released for DocNADEe.

### C.1  EXPERIMENTAL SETUP FOR GENERALIZATION

We set the maximum number of training passes to 100, topics to 200 and the learning rate to 0.001 with *sigmoid* hidden activation. Since the baseline DocNADE and DocNADEe reported better scores in PPL for $H = 200$ topics than using $50$, therefore we use $H = 200$ in our experiments.

See Table 11 for hyperparameters used in generalization task, i.e., computing PPL.

See section C.4 to reproduce scores of Table 1.

### C.2  EXPERIMENTAL SETUP FOR IR TASK

We set the maximum number of training passes to 100, topics to 200 and the learning rate to 0.001 with *tanh* hidden activation. Since the baseline DocNADE and DocNADEe reported better scores in

| Hyperparameter | Search Space |
|---|---|
| retrieval fraction | [0.02] |
| learning rate | [0.001] |
| hidden units, $H$ | [200] |
| activation function ($g$) | tanh |
| iterations | [100] |
| $\lambda^k$ | [1.0, 0.5, 0.1] |
| $\gamma^k$ | [0.1, 0.01, 0.001] |

Table 12: Hyperparameters search in the Information (text) Retrieval task, where $\lambda^k$ and $\gamma^k$ are weights for $k$th source. We use the same grid-search for all the source domains. Hyperparameters in IR task in DocNADE, DocNADEe, LVT, GVT and MVT configurations for 200 topics

| setting | KBs from Source Corpus | Model/ Transfer Type | Scores on Target Corpus (*in sparse-data setting*) | | | | | | | | | | | |
|---|---|---|---|---|---|---|---|---|---|---|---|---|---|---|
| | | | 20NSshort | | | TMNtitle | | | R21578title | | | 20NSsmall | | |
| | | | *PPL* | *COH* | *IR* | *PPL* | *COH* | *IR* | *PPL* | *COH* | *IR* | *PPL* | *COH* | *IR* |
| parameterized | *MST* | LVT | 667 | .661 | .308 | 670 | .730 | .535 | 183 | .716 | .661 | 610 | .440 | .286 |
| | | GVT | 651 | .658 | .285 | 701 | .712 | .523 | 190 | .701 | .656 | 602 | .460 | .273 |
| | | MVT | 667 | .660 | .309 | 667 | .730 | .535 | 183 | .714 | .661 | 608 | .441 | .293 |
| | | + Glove | 662 | .677 | .296 | 672 | .731 | .540 | 183 | .716 | .662 | 634 | .412 | .207 |
| hyper-parameterized | *MST* | LVT | **640** | .678 | .308 | **663** | .732 | .547 | **182** | .739 | .673 | **594** | .542 | .277 |
| | | GVT | 658 | .705 | .305 | 704 | .746 | .550 | 192 | .727 | .673 | 599 | .585 | **.326** |
| | | MVT | 656 | **.740** | **.314** | 680 | **.752** | **.569** | 188 | .745 | **.685** | 600 | **.637** | .285 |

Table 13: $\{\lambda, \gamma\}$ as Parameter vs Hyperparameters: Perplexity (PPL), topic coherence (COH) and precision (IR) at retrieval fraction 0.02, when $\lambda$ and $\gamma$ are (1) learned with backpropagation, and (2) treated as hyperparameters. The experimental results suggest that the second configuration performs better the former. Therefore, in the paper we have reported scores considering $\{\lambda, \gamma\}$ as hyperparameters. + Glove: MVT+Glove embeddings. *Please read column-wise.* **Bold**: best in column.

precision for the retrieval task for $H = 200$ topics than using 50, therefore we use $H = 200$ in our experiments. We follow the similar experimental setup as in DocNADEe. For model selection, we used the validation set as the query set and used the average precision at 0.02 retrieved documents as the performance measure. Note that the labels are not used during training. The class labels are only used to check if the retrieved documents have the same class label as the query document. To perform document retrieval, we use the same (Table 2) train/development/test split of documents for all the datasets during learning.

Given DocNADE, the representation of a document of size $D$ can be computed by taking the last hidden vector $\mathbf{h}_D$ at the autoregressive step $D$. Since, the RSM and DocNADE strictly outperformed LDA, therefore we only compare DocNADE and its recent extension DocNADEe. We use the same number of topic dimensions ($H = 200$) across all the source domains and the target in training with DocNADE.

See Table 12 for the hyperparameters in the document retrieval task, where $\lambda^k$ and $\gamma^k$ are weights for $k$th source. We use the same grid-search for all the source domains. We set $\gamma^k$ smaller than $\lambda^k$ to control the degree of imitation of the source domain(s) by the target domain. We use the development set of the target corpus to find the optimal setting in different configurations of knowledge transfers from several sources.

See section C.4 to reproduce scores of Table 5.

## C.3 $\{\lambda, \gamma\}$ AS PARAMETER VS HYPERPARAMETERS

Here, we treat $\lambda$ and $\gamma$ as parameters of the model, instead of hyperparameters and learn them with backpropagation. We initialize each $\lambda^k = 0.5$ and $\gamma^k = 0.01$ for each of the sources. We perform experiments on short-text datasets in MST+LVT, MST+GVT and MST+MVT configurations. We evaluate the topic modeling using PPL, topic coherence and retrieval accuracy.

Table 13 reports the scores, when $\lambda$ and $\gamma$ are (1) learned with backpropagation, and (2) treated as hyperparameters. The experimental results suggest that the second configuration performs better the former. Therefore in the work, we have reported scores considering $\{\lambda, \gamma\}$ as hyperparameters.

## C.4  REPRODUCIBILITY: OPTIMAL CONFIGURATIONS OF $\lambda$ AND $\gamma$

As mentioned in Tables 11 and 12, the hyper-parameter $\lambda^k$ takes on values in [1.0, 0.5, 0.1] for each of the word embeddings matrix $\mathbf{E}^k$ and $\gamma^k$ in [0.1, 0.01, 0.001] for each of the latent topic features $\mathbf{Z}^k$, respectively for the $k^{th}$ source domain. To determine an optimal configuration, we perform grid-search over the values and use the scores on the development set to determine the best setting. We have a common model for PPL and COH scores due to generalization criteria.

To **reproduce** scores (best in Table 5), we mentioned the best settings of ($\lambda^k$, $\gamma^k$) in MST+MVT configuration for each of the target and source combinations:

1. Generalization (PPL and COH) in MST+MVT when **target** is 20NSshort: ($\lambda^{20NS} = 1.0$, $\gamma^{20NS} = 0.001$, $\lambda^{TMN} = 0.1$, $\gamma^{TMN} = 0.001$, $\lambda^{R21578} = 0.5$, $\gamma^{R21578} = 0.001$, $\lambda^{AGnews} = 0.1$, $\gamma^{AGnews} = 0.001$

2. Generalization (PPL and COH) in MST+MVT when **target** is TMNtitle: ($\lambda^{20NS} = 0.1$, $\gamma^{20NS} = 0.001$, $\lambda^{TMN} = 1.0$, $\gamma^{TMN} = 0.001$, $\lambda^{R21578} = 0.5$, $\gamma^{R21578} = 0.001$, $\lambda^{AGnews} = 1.0$, $\gamma^{AGnews} = 0.001$

3. Generalization (PPL and COH) in MST+MVT when **target** is R21578title: ($\lambda^{20NS} = 0.1$, $\gamma^{20NS} = 0.001$, $\lambda^{TMN} = 0.5$, $\gamma^{TMN} = 0.001$, $\lambda^{R21578} = 1.0$, $\gamma^{R21578} = 0.001$, $\lambda^{AGnews} = 1.0$, $\gamma^{AGnews} = 0.001$

4. Generalization (PPL and COH) in MST+MVT when **target** is 20NSsmall: ($\lambda^{20NS} = 0.5$, $\gamma^{20NS} = 0.001$, $\lambda^{TMN} = 0.1$, $\gamma^{TMN} = 0.001$, $\lambda^{R21578} = 0.1$, $\gamma^{R21578} = 0.001$, $\lambda^{AGnews} = 0.1$, $\gamma^{AGnews} = 0.001$

5. Generalization (PPL and COH) in MST+MVT when **target** is Ohsumedtitle: ($\lambda^{AGnews} = 0.1$, $\gamma^{AGnews} = 0.001$, $\lambda^{PubMed} = 1.0$, $\gamma^{PubMed} = 0.001$

6. Generalization (PPL and COH) in MST+MVT when **target** is Ohsumed: ($\lambda^{AGnews} = 0.1$, $\gamma^{AGnews} = 0.001$, $\lambda^{PubMed} = 1.0$, $\gamma^{PubMed} = 0.001$

7. IR in MST+MVT when **target** is 20NSshort: ($\lambda^{20NS} = 1.0$, $\gamma^{20NS} = 0.1$, $\lambda^{TMN} = 0.5$, $\gamma^{TMN} = 0.01$, $\lambda^{R21578} = 0.1$, $\gamma^{R21578} = 0.001$, $\lambda^{AGnews} = 1.0$, $\gamma^{AGnews} = 0.01$

8. IR in MST+MVT when **target** is TMNtitle: ($\lambda^{20NS} = 0.1$, $\gamma^{20NS} = 0.01$, $\lambda^{TMN} = 1.0$, $\gamma^{TMN} = 0.01$, $\lambda^{R21578} = 0.1$, $\gamma^{R21578} = 0.01$, $\lambda^{AGnews} = 0.5$, $\gamma^{AGnews} = 0.001$

9. IR in MST+MVT when **target** is R21578title: ($\lambda^{20NS} = 0.1$, $\gamma^{20NS} = 0.01$, $\lambda^{TMN} = 1.0$, $\gamma^{TMN} = 0.01$, $\lambda^{R21578} = 1.0$, $\gamma^{R21578} = 0.01$, $\lambda^{AGnews} = 0.5$, $\gamma^{AGnews} = 0.001$

10. IR in MST+GVT when **target** is 20NSsmall: ($\gamma^{20NS} = 0.01$, $\gamma^{TMN} = 0.01$, $\gamma^{R21578} = 0.1$, $\gamma^{AGnews} = 0.01$

11. IR in MST+MVT when **target** is Ohsumedtitle: ($\lambda^{AGnews} = 0.1$, $\gamma^{AGnews} = 0.001$, $\lambda^{PubMed} = 1.0$, $\gamma^{PubMed} = 0.1$

12. IR in MST+MVT when **target** is Ohsumed: ($\lambda^{AGnews} = 0.1$, $\gamma^{AGnews} = 0.001$, $\lambda^{PubMed} = 0.5$, $\gamma^{PubMed} = 0.1$

The hyper-parameters mentioned above also applies to a single source transfer configuration.

Additionally, we have also provided the **code**.

While DocNADEe requires the dimension (i.e., $E$) of word embeddings be the same as the latent topic (i.e., $H$), we first apply a projection on the concatenation of the pretrained word embeddings obtained from several sources and then, introduce the prior knowledge in each of the autoregressive step following DocNADEe. We apply it in configurations where Glove and/or FastText ($E$=300) are employed. IN these settings, we use a single mixture weight $\lambda \in [1.0, 0.5, 0.1]$ over the projected vector and then, introduced in TM following DocNADEe.

## C.5 EXPERIMENTAL SETUP FOR NVDM AND PRODLDA

For NVDM, we run the code availale at `github.com/ysmiao/nvdm` and train for 200 topics.

For ProdLDA, we run the code availale at `github.com/akashgit/autoencoding_vi_for_topic_models` and train for 200 topics.

## C.6 EXPERIMENTAL SETUP FOR GLOVE-DMM

We used LFTM (`https://github.com/datquocnguyen/LFTM`) to train glove-DMM model. It is trained for 200 iterations with 2000 initial iterations using 200 topics. For short texts, we set the hyperparameter beta to 0.1, for long texts to 0.01; the mixture parameter lambda was set to 0.6 for all datasets. IR task was performed using relative topic proportions as input, where we inferred the topic distribution of the training and test documents and used the relative distribution as input in computing similarities in documents based on the inferred relative topic distribution.

## C.7 EXPERIMENTAL SETUP FOR DOC2VEC

We used gensim (`https://github.com/RaRe-Technologies/gensim`) to train Doc2Vec models. Models were trained with distributed bag of words, for 1000 iterations using a window size of 5 and a vector size of 500.

