# OpenReview forum: "Multi-source Multi-view Transfer Learning in Neural Topic Modeling with Pretrained Topic and Word Embeddings"
_ICLR.cc/2020/Conference — Reject_

### Official Review · AnonReviewer2 · 2019-10-23
**Official Blind Review #2**

**Rating:** 6

**Review:**

On the basis of existing topic modelling approaches, the authors apply a transfer learning approach to incorporate additional knowledge to topic models, using both word embeddings and topic models. The underlying idea is that topic models contain a global view that differs on a thematic level, while word embeddings contain a local, immediate contextual view. The combination of both local and global view transfer to enhance a topic model is the main contribution of this paper, especially when using multiple sources (therefore the title: multi-source multi-view transfer).
Given a document collection, DocNADE is used to generate the topic-word matrix. In the local view transfer step, the pre-trained WordPool is used, from which knowledge is transferred on the target document. The global view transfer is done by transferring knowledge from the pre-trained TopicPool to the target. As described in Algorithm 1 in the paper, both Word- and TopicPool are jointly used in the transfer learning process.
For evaluation, three different measures are taken into account: Perplexity, Topic Coherence and Precision (Information Retrieval). In comparison to a DocNADE only approach, all values are better in the settings that use the transfer learning approach. Compared to DocNADE + word embeddings, the results are competitive as well. In both experiments, the multi-source setting evaluates best overall.

In conclusion, the paper shows that exploiting multiple sources and views in transfer learning leads to an overall improvement in the given tasks. The main contribution is the usage topic models in a transfer learning framework. Additionally the use of multi-source word embeddings is novel too, especially in the joint setting with the topic model transfer. The paper shows how the DocNADE approach is enhanced to make use of both local and global view transfer and how this enhancement leads to improved performance on various related tasks.
Still, the overall contribution is mostly in combining existing methods and can be judged as rather incremental.

Minor note: A small mistake has been found in Table 5. The best perplexity value in the first column is not the bold 638, but the 630 in the local-view transfer setting.

Edit after rebuttal: In my review I did not value the contribution of the transfer learning approach enough. So, when also considering the extensive evaluation I am now leaning towards accept.

**Experience Assessment:**

I have published one or two papers in this area.

**Review Assessment: Checking Correctness Of Derivations And Theory:**

I assessed the sensibility of the derivations and theory.

**Review Assessment: Checking Correctness Of Experiments:**

I assessed the sensibility of the experiments.

**Review Assessment: Thoroughness In Paper Reading:**

I read the paper at least twice and used my best judgement in assessing the paper.

---

> ### Author Response · Authors · 2019-11-07
> **About "combining existing methods and can be incremental": First Work to perform Multi-view and Multi-source transfer learning in Neural topic modeling**
>
> Thanks for your reviews, positive comments about "novelty" and acknowledging gains obtained by our proposed modeling.
>
> As far as we know, this is the first/novel work that introduces:
> (1) single-source pre-trained topic embeddings,
> (2) single-source joint pre-trained word and topic embeddings, and
> (3) multi-source transfers using pre-trained topics and word embeddings jointly in neural topic modeling under transfer learning paradigm.
>
> This work DOES NOT focus on introducing a new topic model; however, we focus on introducing a novel transfer learning mechanism in neural topic modeling using complementary representations. Therefore, we have used the existing neural topic model, i.e., DocNADE to address data sparsity issues.
>
> The experimental results have clearly shown noticeable gains in topic modeling due to the proposed transfer learning methodology using 7 target datasets from several domains (e.g., news, medical, etc.), evaluated using perplexity, topic coherence and information retrieval task.
>
> Thanks for the minor comment. We will correct it and will update the gain% as well. :)

---

> ### Author Response · Authors · 2019-11-15
> **"Enough contribution of transfer learning"**
>
> Thanks for increasing your rating and leaning towards accept!
>
> Thanks for acknowledging contribution of our proposed transfer learning approaches in topic modeling.

---

### Official Review · AnonReviewer3 · 2019-10-30
**Official Blind Review #3**

**Rating:** 3

**Review:**

The paper proposes a multi-source and multi-view transfer learning for neural topic modelling with the pre-trained topic and word embedding. The method is based on NEURAL AUTOREGRESSIVE TOPIC MODELs --- DocNADE (Larochelle&Lauly,2012). DocNADE learns topics using language modelling framework. DocNADEe (Gupta et al., 2019) extended DocNADE by incorporating word embeddings, the approach the authors described as a single source extension of the existing method.

In this paper, the proposed method adds a regularizer term to the DocNADE loss function to minimize the overall loss whereas keeping the existing single-source extension. The authors claimed that incorporating the regularizer will facilitate learning the (latent) topic features in the trainable parameters simultaneously and inherit relevant topical features from each of the source domains and generate meaningful representations for the target domain. The analysis and evaluation were presented to show the effectiveness of the proposed method. However, the results are not significantly improved than the based line model DocNADE.

Overall, the paper is written well. However, it is not clear to me that the improved results are resulted due to multi-source multi-view transfer learning or for the better leaning of the single-source model due to the incorporation of the regularizer.




**Experience Assessment:**

I have published one or two papers in this area.

**Review Assessment: Checking Correctness Of Derivations And Theory:**

I did not assess the derivations or theory.

**Review Assessment: Checking Correctness Of Experiments:**

I did not assess the experiments.

**Review Assessment: Thoroughness In Paper Reading:**

I read the paper at least twice and used my best judgement in assessing the paper.

---

> ### Author Response · Authors · 2019-11-07
> **"20 Evidences of significant improvements using 7 datasets (small/large) across 3 evaluation measures"**
>
> Thanks for your reviews and positive comments, e.g., "well written".
>
> The extensive experimental results (Table 5, 6 and 7) have shown significant improvements in terms of perplexity (PPL), topic coherence (COH) and IR scores using 7 datasets.  The improvements are EXPLICITLY mentioned in Tables 5, 6 and 7 (see "Gain%"). Also, see plots 2 (a,b,c,d,e), where our proposed model outperforms all the baselines at all the fractions in terms of retrieval precision.
>
> Beyond perplexity, we have also shown large gains in topic coherence scores due to improved topic quality and noticeable gains in precision for IR task.
>
> Following are the 20 (some) EVIDENCES of significant improvements:
>
> "Gain% vs DocNADE baseline" (Table 5):
> On 20NSshort: 10.9% (COH), 8.28% (IR)
> On TMNtitle: 7.22% (PPL), 6.06% (COH) and 9.21% (IR)
> On 20NSsmall: 37.9% (COH), 20.7% (IR).
> On Ohsumedtitle: 4.01% (PPL) and 13.8% (IR) (Table 7)
> On Ohsumed: 12.3% (PPL) and 4.35% (IR) (Table 7)
>
>
> "Gain% vs DocNADEe baseline" (Table 6):
> On 20NSshort: 9.95% (COH), 8.84% (IR)
> On TMNtitle: 4.60% (COH) and 7.04% (IR)
> On 20NSsmall: 39.3% (COH).
> On Ohsumedtitle: 17.3% (PPL) and 4.0% (IR) (Table 7)
> On Ohsumed: 8.5% (PPL) and 4.91% (IR) (Table 7)
>
> Additionally, #R4 and #R2 have acknowledged the noticeable gains achieved in this paper.

---

> ### Author Response · Authors · 2019-11-07
> **About the "regulariser and clear contribution of each component"**
>
> As far we we know, we have covered all the experimental settings, where we have clearly/individually shown contributions of each of the components.
>
> See Table 5, 6 and 7, where the scores are reported due to:
> (1) only single-source word embedding transfer, i.e., LVT,
> (2) multi-source word embedding transfer, i.e., MST+LVT,
> (3) only single-source topic embedding transfer, i.e., GVT,
> (4) multi-source topic embedding transfer, i.e., MST+GVT,
> (5) single-source joint word and topic embeddings transfers, i.e., MVT=LVT+GVT, and
> (6) multi-source joint word and topic embeddings transfers, i.e., MST+ MVT.
>
> Notice that the topic-embedding transfer is performed via the regulalrization term. Also, mentioned in algorithm #1.
>
> We are happy to answer if something is still not clear. Please point out precisely.

---

### Official Review · AnonReviewer4 · 2019-11-04
**Official Blind Review #4**

**Rating:** 6

**Review:**

This is an emergency review.

This work proposes a novel method to use pre-trained topic embeddings and pre-trained word embeddings obtained from various corpora in the transfer learning framework.

Their model architecture is based on DocNADE, unsupervised neural-network based topic model, and the authors propose two strategies to use pre-trained topic embeddings and pre-trained word vectors.
1) Addition of a weighted sum of pre-trained word embeddings and the hidden vector of DocNADE.
2) L2-Regularization term between topic embedding of DocNADE and pre-trained topic embeddings. They propose to align these two embeddings by multiplying align matrix "A" to the topic embedding of DocNADE.

They show the transfer learning performance of their model on various source/target domain datasets, including medical target corpora, and verify that their model outperforms on a short text and small document collection.

Strengths.
1. Comparison with the data augmentation baseline shows the performance gain is not only from bigger training data. Even though comparison with the naive baseline (data augmentation) seems too obvious, I think the results clearly show their claim about the importance of using transfer learning in neural topic modeling domain.
2. As the first approach that introduces a novel transfer learning framework with pre-trained topic embeddings, they show tons of experimental results with various datasets and metrics to show the specification of their method. Their experimental setting is well designed.

Weaknesses and comments:
Their method to combine pre-trained word embeddings and pre-trained topic embeddings is too simple. Since this is the first approach to use topic embedding in the transfer learning field, the simplicity of the proposed method is somewhat necessary. However, a weighted sum of pre-trained topic/word vectors seems not enough to transfer multisource knowledge. For instance, word vectors obtained from individual training processes do not share embedding vector space. As you apply the alignment method to topic embeddings from various sources, you should align word embeddings too.

**Experience Assessment:**

I have read many papers in this area.

**Review Assessment: Checking Correctness Of Derivations And Theory:**

I assessed the sensibility of the derivations and theory.

**Review Assessment: Checking Correctness Of Experiments:**

I assessed the sensibility of the experiments.

**Review Assessment: Thoroughness In Paper Reading:**

I read the paper at least twice and used my best judgement in assessing the paper.

---

> ### Author Response · Authors · 2019-11-07
> **About "Word Embedding Alignment": Yes, we do align Word Embeddings (mentioned 3 times in the paper)**
>
> Thanks for your (emergency) reviews.
>
> Thanks for your positive comments on experimental setup and acknowledging that our transfer learning approaches introduced in neural topic modeling clearly outperform several baselines.
>
> >> "Word Embedding Alignment"
> Yes, we do.
> Please see section 3, page 6 in "Reproducibility" paragraph (line 3). Also, mentioned in caption of figure 6 as well as in Appendix C.4 (the last paragraph).
>
> We perform the word embeddings alignment in all the "+Glove" settings (Table 6) to
> (1) overcome the DocNADEe (baseline topic model) limitation (word-embedding size must be same as the number of topics), and
> (2) align vector spaces of word-embeddings obtained from several sources as well as from several different training processes, e.g., from Glove, FastText and word embeddings from topic models.
>
> The focus of our work is to demonstrate the joint word and topic embeddings transfer in neural topic models from one or many sources.

---

### Author Response · Authors · 2019-11-12
**Rejection without a single constructive/negative comment? Please justify negative scores!**

Dear Reviewers,

Thanks again for reviewing our paper! We have responded to your queries and we are looking forward to discuss further.

Even though there is NO negative/critical criticism, the ratings are NOT positive. We mostly found clarification queries that we have addressed in our response. We have also highlighted our contributions and SIGNIFICANT GAINS that our proposed methods achieved.

We would appreciate if the reviewers could participate in the rebuttal and raise further questions, if any.

Also, we would acknowledge if you could justify your negative ratings or update them accordingly based on our response below.

Thanks!

---

### Decision · Program_Chairs · 2019-12-19

**Decision:**

Reject

**Comment:**

This paper presents a transfer learning framework in neural topic modeling. Authors claim and reviewers agree that this view of transfer learning in the realm of topic modeling is novel.

However, after much deliberation and discussion among the reviewers, we conclude that this paper does not contribute sufficient novelty in terms of the method. Also, reviewers find the experiments and results not sufficiently convincing.

I sincerely thank the authors for submitting to ICLR and hope to see a revised paper in a future venue.